# An evolutionary analysis of supply chain collaborative information sharing based on prospect theory

**Meng Liu[1,2], Luyu Zhai[3], Hongcheng Gan** [iD][1,2] *

1 Business School, University of Shanghai for Science and Technology, Shanghai, China, 2 Center for Supernetworks Research, University of Shanghai for Science and Technology, Shanghai, China, 3 School of Economics & Management, University of Science &Technology Beijing, Beijing, China

* 221420098@st.usst.edu.cn

## Abstract

In order to delve into the dynamic evolution process and influencing factors of information sharing decisions among stakeholders under supply chain collaboration, this study constructs an evolutionary game model with suppliers and retailers as the primary entities. Within this model, a combined approach of game theory and prospect theory is employed, integrating prospect value functions and weight functions to create an information sharing prospect value matrix. A comprehensive analysis is conducted on the strategic choices and benefits of entities considering the psychological perception of information sharing, and critical factors influencing the stability of information sharing evolution results are explored through numerical simulations using Matlab. The key findings of this study are as follows: Firstly, from the perspective of supply chain collaboration, the probability of entities evolving into information sharing is negatively correlated with the cost of information sharing and positively correlated with the benefits generated by information coordination. Secondly, looking at supply chain collaboration, entities are more likely to engage in information sharing behavior when they exhibit a lower level of risk aversion, indicating greater rationality, when facing profits; conversely, they are more likely to participate in information sharing when they display a higher degree of risk preference, indicating less rationality, in the face of losses. Furthermore, the lesser sensitivity of suppliers and retailers to losses is more likely to drive the system towards an information-sharing state. Based on the primary findings mentioned above, this study offers recommendations for enhancing trust, constructing information exchange platforms, and adjusting psychological awareness. These suggestions contribute to improving information sharing among entities within the supply chain, thus enhancing the overall efficiency and collaboration of the supply chain.

## 1. Introduction

In light of the shifting market landscape and intensifying global competition, supply chain collaboration has emerged as a pivotal means for enterprises to enhance their competitiveness

**Data Availability Statement:** All relevant data are in the manuscript and its Supporting Information files.

**Funding:** The author(s) received no specific funding for this work.

**Competing interests:** The authors have declared that no competing interests exist.

and operational efficiency [1, 2]. It has progressively evolved into a central and critical topic within modern supply chain management [3]. Its primary objective is to elevate the efficiency and benefits of the entire supply chain through effective information sharing and collaborative decision-making [4, 5]. Information sharing among supply chain members can effectively alleviate the bullwhip effect and double marginalization effect caused by asymmetric information among members [6, 7], thereby further achieving supply chain collaboration [8], thus enhancing the overall competitiveness of the supply chain. Effective information sharing mechanisms can help supply chains achieve supply chain collaboration and stability, which is currently the goal of supply chain management.

However, in practice, supply chain information sharing still faces numerous challenges, such as willingness to share information, asymmetric information [9], information matching, and varying abilities of shared entities to absorb and utilize information [10]. This complexity necessitates critical deliberation among supply chain entities regarding whether to engage in information sharing. To explore the underlying decision-making mechanisms, this study constructs a game model composed of suppliers and retailers. The aim is to investigate how various factors influence the information sharing decisions of these entities.

In recent years, numerous scholars have analyzed and summarized the information sharing behavior from the perspective of supply chains. Sheng has constructed an information sharing model based on blockchain to solve the asymmetric information in the supply chain transaction process [7]. Dan et al. [11] focusing on fresh agricultural products supply chain, explored the optimal pricing and preservation efforts under different information sharing strategies and studied the information sharing strategies of retailers. Liao et al. [12] studied a secondary supply chain consisting of manufacturers with production capacity constraints and two retail markets with differing demand, exploring the value creation and willingness to share demand information in the presence of capacity constraints and various allocation methods. These studies are mostly based on the "rational economic man" hypothesis of conventional economic theory, which assumes that people are completely rational, selfish, and pursue maximum benefits. However, with the rise of behavioral economics, people have gradually realized the shortcomings of the "rational economic man" hypothesis in traditional economics [13]. Behavioral economics believes that people's psychological factors cannot be ignored when making decisions. Many studies have also shown that the psychological factors of supply chain members will affect their decision-making and the overall efficiency of the supply chain, and their behavior is significantly different from the predicted results of the standard economic model [14, 15]. Therefore, the psychological perception of the subjects is an important behavior factor that cannot be ignored in the study of supply chains, and it is urgent to consider the subjective psychological factors of the subjects in supply chain information sharing.

At present, many scholars have incorporated the subjective psychological perception factor of the subject into the behavior decision-making of supply chain members. Numerous scholars observed that supply chain members have a great concern for fairness, namely, fairness concern [16–18]. Scholars believe that under the influence of fairness concern, people are likely to take action at the cost of their own interests to punish the other party when they feel unfair. Fairness concern is inconsistent with the traditional expected utility theory, which violates the rational person hypothesis. Many empirical or experimental studies have confirmed the existence of this behavioral tendency [19]. The above research is mostly based on psychological considerations of gains. In other words, the above research is mostly based on the subject's psychological perception of "gain". However, prospect theory [20] tells us that losing has a greater impact on psychology than gaining, and people generally have a mentality of loss aversion [21, 22]. Given the greater influence of "loss" on psychological perception, scholars consider loss aversion. Wang et al. [23] based on the problem of supply chain financing strategy

under random demand situations, considering the impact of loss aversion of fund-constrained retailers on supply chain operation decision-making and financing strategies. Qiu [24] analyzed the impact of consumer service loss aversion on enterprise pricing decisions and profits in the case of jointly building a multi-channel supply chain by manufacturers and retailers. The results show that the service loss aversion of consumers leads to a decrease in retail prices, online sales prices, and wholesale prices. Chen et al. [25] studied the ordering decision-making problem of loss-averse salespeople under option contracts and analyzed in detail the impact of parameters such as loss aversion coefficient of salespeople, retail price, and execution price on their ordering decisions.

Compared to the above literatures, the difference in this paper is that the literatures mostly study the subject's information sharing behavior decision-making from the perspective of the supply chain and less from the perspective of supply chain collaboration. The difference between the two is that information sharing in the supply chain is only a conventional form of supply chain cooperation. At present, with the changing market environment and the increasing global competition, supply chain collaboration has become an important means to improve enterprise competitiveness and operational efficiency. In previous research, some scholars have also contended that trust mechanisms, collaborative decision-making, and supply chain visibility are vital determinants of supply chain collaboration. Nevertheless, information sharing has often been considered as the fundamental prerequisite for supply chain collaboration in numerous studies, sometimes even synonymous with collaboration itself. Consequently, in the interest of model simplification, this study exclusively investigates scenarios where supply chain collaboration is achieved through information sharing alone. Achieving supply chain collaboration through information sharing has become an important goal of supply chain management at present. Viewing information sharing behavior from the perspective of supply chain collaboration will be more in line with the current research background, but it also increases the difficulty of decision-making. Moreover, at present, there are relatively few studies on the considerations of gains and losses that decision-making subjects feel when sharing information from the perspective of supply chain collaboration, and there is a lack of relevant quantitative research.

Based on the above business observations and literatures review, this study will mainly explore the following issues: 1. From the perspective of supply chain collaboration, explore the influence of various factors on information sharing behavior decision-making; 2. Considering that the behavior characteristics of supply chain subjects are not completely rational, integrate prospect theory into the analysis to assess how the psychological perception of information sharing subjects regarding gains and losses influences their decisions.

In light of the aforementioned limitations, this study, grounded in the perspective of supply chain collaboration, seeks to uncover the latent psychological and behavioral factors in supply chain information sharing decision-making. Specifically, drawing from the assumption of bounded rationality in game theory [26] and combining it with prospect theory, the study examines how the perception of value and risk aversion by game players influence the patterns of system evolution and stability. Through these investigations, we aim to provide a deeper understanding for enterprises and supply chain practitioners, enabling them to refine their decision-making and practices. Furthermore, we hope this research contributes to enriching the foundational theoretical framework of supply chain collaboration and offers new perspectives and scientific methods to researchers in the field.

The main contributions of this study are as follows: 1. It analyzes the behavioral decisions of supply chain decision-makers from the perspective of supply chain collaboration, offering a novel viewpoint. 2. By introducing prospect theory into the domain of supply chain collaboration management, the study provides a fresh theoretical framework that elucidates the

evolutionary process of information sharing in supply chains, enriching the existing supply chain theory. 3. This research can assist supply chain decision-makers in better understanding the impact of psychological factors on their decision-making, providing effective guidance for practice. 4. The study also presents research insights and future prospects, serving as a reference and offering suggestions for related researchers.

## 2. Information sharing evolution model under supply chain collaboration

### 2.1 Model assumptions

1. This study aims to analyze the information sharing behavior among supply chain partners from the perspective of supply chain collaboration. To achieve this, a two-player evolutionary game model based on game theory will be constructed. The rationale for this is as follows: 1. Game theory is an intuitive and effective tool in economics for solving problems related to behavioral choice of agents; 2. Agents engaging in information sharing behavior make choices based on limited information and their own costs; 3. The interests of these entities are influenced by their behavioral choices, aligning with the typical characteristics of game theory [27]. In this study, we present a simplified model, assuming a two-tier supply chain comprising suppliers and retailers with bounded rationality. Since the two parties are influenced by different environments and corporate cultures, they are under conditions of incomplete information when making decisions. Additionally, their bounded rationality means that they will gradually move towards the optimal state through continuous learning and strategy adjustment. Furthermore, both have only two choices: information sharing or non-sharing.

2. The information shared among the entities in the supply chain encompasses elements such as market demand forecasting, inventory data, as well as transportation and logistics information, all of which contribute to optimizing decision-making for both parties. However, it does not include information concerning each entity's core competitive advantages. Moreover, the act of information exchange between these entities is imbued with numerous uncertainties, including factors like the market environment, uncertainties about the long-term benefits of information sharing, and the costs associated with information maintenance. When making decisions within the context of this game, the agents base their choices not on anticipated utility values, but rather on their own perceptions of how strategies ultimately generate value. According to the cumulative prospect theory [20] proposed by Amos and Daniel, the perceived value of the information sharing entities can be measured by the prospect value $V$, which is determined by the value function $v(\Delta x)$ and weight function $w(p)$, that is $V = \sum v(\Delta x)w(p)$. Substituting Prospect Theory function for the expected utility function aligns more closely with reality.

$$w(p_i) = \frac{p_i^{\gamma}}{(p_i^{\gamma} + (1 - p_i)^{\lambda})^{\frac{1}{\gamma}}}$$

$$v(\Delta x) = \begin{cases} (\Delta x)^{\alpha}, & \Delta x \geq 0 \\ -\lambda(-\Delta x)^{\beta}, & \Delta x < 0 \end{cases}$$

where $p$ represents the objective probability that $i$ occurs. The weight function $w(p_i)$ reflects the impact of $p_i$ on the overall effect, which has a shape of inverted "$S$", and the larger the $\gamma$

value, the smaller the curvature of the function curve. In general, people assign a weight of 1 to events with extremely high probability and assign a weight of 0 to events with extremely low probability, that is $w(1) = 1$ and $w(0) = 0$. We tend to underestimate events with medium and high probability and overestimate events with low probability in the actual decision-making process. $\Delta x_i$ is the difference between the actual benefit of the decision-making group and the reference point after the decision-making event occurs. $v(\Delta x)$ refers to the subjective feeling value of the decision-making subject after the decision-making event occurs. Assuming that the supplier and the retailer exhibit risk aversion degrees $\alpha_1$, $\alpha_2 \in (0,1)$ when facing gains and risk preference degree $\beta_1, \beta_2 \in (0,1)$ when confronting losses. $\lambda(\lambda \geq 1)$ denotes the loss aversion coefficient, where a higher value signifies that the game agents are more sensitive to losses compared to gains. Additionally, we posit that one party's information sharing strategy will not influence the psychological state of the other party. The reference point selection is also very important, because it is used to judge the gains and losses of the decision-making subject. In this study, the benefits obtained when all supply chain information sharing subjects choose not to share information are taken as the reference point; that is, the perceived value is 0.

3. Literatures [7, 11, 28] are used as key references for parameter assumptions. We assume that the initial gains for both to be 0 when neither the supplier nor the retailer shares information. Suppliers and retailers are participants, and their respective amount of information sharing are $b_1$ and $b_2$. Since this study exclusively examines cases where supply chain collaboration is achieved through information sharing, we assume that when both parties engage in information sharing, suppliers and retailers attain collaboration through information sharing and generate collaborative gains [29]. However, scholars only assume the collaborative benefit as a fixed value. In contrast, this study contends that the magnitude of collaborative gains depends on the complementarity and volume of shared information. We assume that the complementarity is $t_1$ and $t_2$ respectively, and the amount of information sharing is $b_1$ and $b_2$ respectively, the collaborative benefits obtained by both parties are $t_1 b_1$ and $t_2 b_2$ respectively. However, suppliers and retailers cannot fully absorb the received information when they are receivers [30]. Assume that their respective information absorption and conversion capability coefficients are $e_1$ and $e_2$, and information sharing costs are $c_1$ and $c_2$. The cost refers to the information investment cost, information security cost, and information maintenance cost of the database or information system for the connection and upgrading when the participants share information [31]. Assume that the additional benefits brought about by long-term information sharing between supplier and retailer (such as increased reputation in the industry and increased market share) are $m$ and $n$.

4. (In the game, the probability that the supplier chooses information sharing is $x(0 \leq x \leq 1)$, and the probability for choosing the information non-sharing is $1-x$; the probability that the retailer chooses information sharing is $y(0 \leq y \leq 1)$, and the probability for choosing the information non-sharing is $1-y$.

## 2.2 Model building and solution

Based on the above assumptions regarding collaborative information sharing in supply chains, this study constructs the following game matrix for the information sharing problem within the supply chain collaboration context. Combining this with prospect theory, we obtain the prospect matrix for information sharing, as shown in Table 1 below.

Based on the values in the table, the following calculations can be performed:

**Table 1. Information sharing prospect matrix.**

| | | Retailer | |
|---|---|---|---|
| | | **Information sharing $y$** | **Information non-sharing $1-y$** |
| Supplier | Information sharing $x$ | $m^{\alpha_1} - \lambda c_1^{\beta_1} + (e_1 b_2)^{\alpha_1} + (t_1 b_2)^{\alpha_1},$ $n^{\alpha_2} - \lambda c_2^{\beta_2} + (e_2 b_1)^{\alpha_2} + (t_2 b_1)^{\alpha_2}$ | $\lambda c_1^{\beta_1}, (e_2 b_1)^{\alpha_2}$ |
| | Information non-sharing $1-x$ | $(e_1 b_2)^{\alpha_1}, -\lambda c_2^{\beta_2}$ | 0, 0 |

The expected prospect value and the average prospect value of the supplier under the two strategies of information sharing and information non-sharing are respectively:

$$U_1 = y[m^{\alpha_1} - \lambda c_1^{\beta_1} + (e_1 b_2)^{\alpha_1} + (t_1 b_2)^{\alpha_1}] + (1-y)(-\lambda c_1^{\beta_1}) \tag{1}$$

$$U_2 = y(e_1 b_2)^{\alpha_1} + (1-y) \cdot 0 \tag{2}$$

$$U_x = xU_1 + (1-x)U_2 \tag{3}$$

The dynamic replication equation for the suppliers is:

$$F(x) = \frac{dx}{dt} = x(U_1 - U_x) = x(1-x)[ym^{\alpha_1} + y(t_1 b_2)^{\alpha_1} - \lambda c_1^{\beta_1}] \tag{4}$$

Similarly, the expected prospect value and the average prospect value of the retailer under the two strategies of information sharing and information non-sharing are respectively:

$$U_3 = x[n^{\alpha_2} - \lambda c_2^{\beta_2} + (e_2 b_1)^{\alpha_2} + (t_2 b_1)^{\alpha_2}] + (1-x)(-\lambda c_2^{\beta_2}) \tag{5}$$

$$U_4 = x(e_2 b_1)^{\alpha_2} + (1-x) \cdot 0 \tag{6}$$

$$U_y = yU_3 + (1-y)U_4 \tag{7}$$

The dynamic replication equation for the retailer is:

$$F(y) = \frac{dy}{dt} = y(U_3 - U_y) = y(1-y)[xn^{\alpha_2} + x(t_2 b_1)^{\alpha_2} - \lambda c_2^{\beta_2}] \tag{8}$$

Make these two dynamic replication equations a simultaneous equation and set $F(x) = 0$ and $F(y) = 0$, the equilibrium points of the game system are obtained as $O(0,0)$, $A(1,0)$, $B(0,1)$, $C(1,1)$ and $D(x^*,y^*)$. If and only if $0 \leq \frac{\lambda c_2^{\beta_2}}{n^{\alpha_2} + (t_2 b_1)^{\alpha_2}} = x^* \leq 1$ and $0 \leq \frac{\lambda c_1^{\beta_1}}{m^{\alpha_1} + (t_1 b_2)^{\alpha_1}} = y^* \leq 1$ hold, $D(x^*,y^*)$ is also the equilibrium point of the game system.

## 2.3 Stability analysis of equilibrium point

It can be concluded from the above results that there are five possible local equilibrium points of the game in the system: $O(0,0)$, $A(1,0)$, $B(0,1)$, $C(1,1)$ and $D(x^*,y^*)$. In order to determine the evolutionary stable strategy point of the system, it is necessary to determine the local equilibrium point. Friedman proposed that when $det(J)>0$ and $tr(J)<0$ hold at the same time, the equilibrium point will gradually tend to the local stable state of the system, and the signs of the two can be used to judge whether the system is in an evolutionary stable state. According to the group dynamics of the supplier and the retailer, the Jacobian matrix $J$ is $J = \begin{bmatrix} \frac{\partial F(x)}{\partial x} & \frac{\partial F(x)}{\partial y} \\ \frac{\partial F(y)}{\partial x} & \frac{\partial F(y)}{\partial y} \end{bmatrix}$, where

$\frac{\partial F(x)}{\partial x} = (1 - 2x)[ym^{\alpha_1} + y(t_1 b_2)^{\alpha_1} - \lambda c_1^{\beta_1}], \frac{\partial F(x)}{\partial y} = x(1 - x)[m^{\alpha_1} + (t_1 b_2)^{\alpha_1}]$

$\frac{\partial F(y)}{\partial x} = y(1 - y)[n^{\alpha_2} + (t_2 b_1)^{\alpha_2}], \frac{\partial F(y)}{\partial y} = (1 - 2y)[xn^{\alpha_2} + x(t_2 b_1)^{\alpha_2} - \lambda c_2^{\beta_2}]$ According to the Jacobian matrix, the corresponding determinant and trace are:

$$\det(J) = (1 - 2x)(1 - 2y)[ym^{\alpha_1} + y(t_1 b_2)^{\alpha_1} - \lambda c_1^{\beta_1}][xn^{\alpha_2} + x(t_2 b_1)^{\alpha_2} - \lambda c_2^{\beta_2}]$$
$$- xy(1 - x)(1 - y)[m^{\alpha_1} + (t_1 b_2)^{\alpha_1}][n^{\alpha_2} + (t_2 b_1)^{\alpha_2}] \tag{9}$$

$$tr(J) = (1 - 2x)[ym^{\alpha_1} + y(t_1 b_2)^{\alpha_1} - \lambda c_1^{\beta_1}] + (1 - 2y)[xn^{\alpha_2} + x(t_2 b_1)^{\alpha_2} - \lambda c_2^{\beta_2}] \tag{10}$$

The determinant and trace of the Jacobian matrix corresponding to the calculation system at the local equilibrium point $O(0,0)$, $A(1,0)$, $B(0,1)$, $C(1,1)$ and $D(x^*,y^*)$ are shown in Table 2 below.

It can be seen from Table 2 that the stability of the system is related to many parameters, and the system will be affected by the different values of each parameter. The local stability of the system has the following four cases, and the judgment results of the system equilibrium point in each case are shown in Table 3.

## 2.4 Analysis of evolution results

According to the analysis of Jacobian matrix above, the evolution dynamic phase diagram of the information sharing game between supplier and retailer in different cases (Fig 1). It can clearly show the evolutionary stability strategy of the system in different cases.

In case 1, there are two evolutionary stable strategy points in the system, which are $O(0,0)$ and $C(1,1)$, and in cases 2–4, there is only one evolutionary stable strategy point in the system: $O(0,0)$.

In order to explore the influencing factors of the evolutionary stable state in case 1, the evolution dynamic phase diagram of case 1 was analyzed. In Case 1, the plane is divided into two parts by the broken line ADB (the critical line where the system converges to different states). The system will converge to a state of (sharing, sharing) in the ADBC portion and converge to a state of (non-sharing, non-sharing) in the ADBO portion. According to the set probability, the probability that the system converges to the state of (sharing, sharing) is:

$$P = (1 - x^*)(1 - y^*) + \frac{x^*(1 - y^*)}{2} + \frac{y^*(1 - x^*)}{2} = 1 - \frac{x^*}{2} - \frac{y^*}{2}$$

$$= 1 - \frac{\lambda c_2^{\beta_2}}{2[n^{\alpha_2} + (t_2 b_1)^{\alpha_2}]} - \frac{\lambda c_1^{\beta_1}}{2[m^{\alpha_1} + (t_1 b_2)^{\alpha_1}]} \tag{11}$$

**Table 2. Jacobian matrix of the game model for the supplier and the retailer.**

| Equilibrium point | $\det(J)$ | $tr(J)$ |
|---|---|---|
| $O(0,0)$ | $\lambda^2 c_1^{\beta_1} c_2^{\beta_2}$ | $-\lambda c_1^{\beta_1} - \lambda c_2^{\beta_2}$ |
| $A(1,0)$ | $\lambda c_1^{\beta_1}[n^{\alpha_2} + (t_2 b_1)^{\alpha_2} - \lambda c_2^{\beta_2}]$ | $\lambda c_1^{\beta_1} + [n^{\alpha_2} + (t_2 b_1)^{\alpha_2} - \lambda c_2^{\beta_2}]$ |
| $B(0,1)$ | $[m^{\alpha_1} + (t_1 b_2)^{\alpha_1} - \lambda c_1^{\beta_1}]\lambda c_2^{\beta_2}$ | $[m^{\alpha_1} + (t_1 b_2)^{\alpha_1} - \lambda c_1^{\beta_1}] + \lambda c_2^{\beta_2}$ |
| $C(1,1)$ | $[m^{\alpha_1} + (t_1 b_2)^{\alpha_1} - \lambda c_1^{\beta_1}][n^{\alpha_2} + (t_2 b_1)^{\alpha_2} - \lambda c_2^{\beta_2}]$ | $-[m^{\alpha_1} + (t_1 b_2)^{\alpha_1} - \lambda c_1^{\beta_1}] - [n^{\alpha_2} + (t_2 b_1)^{\alpha_2} - \lambda c_2^{\beta_2}]$ |
| $D(x^*,y^*)$ | $-\lambda^2 c_1^{\beta_1} c_2^{\beta_2} \frac{[m^{\alpha_1} + (t_1 b_2)^{\alpha_1} - \lambda c_1^{\beta_1}][n^{\alpha_2} + (t_2 b_1)^{\alpha_2} - \lambda c_2^{\beta_2}]}{[m^{\alpha_1} + (t_1 b_2)^{\alpha_1}][n^{\alpha_2} + (t_2 b_1)^{\alpha_2}]}$ | $0$ |

**Table 3. Stability analysis of equilibrium point in each case.**

| Case | Restrictions | Equilibrium point | O(0,0) | A(1,0) | B(0,1) | C(1,1) | D(x*,y*) |
|---|---|---|---|---|---|---|---|
| 1 | $m^{\alpha_1} + (t_1 b_2)^{\alpha_1} > \lambda c_1^{\beta_1}$ $n^{\alpha_2} + (t_2 b_1)^{\alpha_2} > \lambda c_2^{\beta_2}$ | det(J) | + | + | + | + | + |
| | | tr(J) | - | + | + | - | 0 |
| | | Stability | ESS | Unstable point | Unstable point | ESS | Saddle point |
| 2 | $m^{\alpha_1} + (t_1 b_2)^{\alpha_1} > \lambda c_1^{\beta_1}$ $n^{\alpha_2} + (t_2 b_1)^{\alpha_2} < \lambda c_2^{\beta_2}$ | det(J) | + | - | + | - | + |
| | | tr(J) | - | +/- | + | +/- | 0 |
| | | Stability | ESS | Saddle point | Unstable point | Saddle point | Disequilibrium point |
| 3 | $m^{\alpha_1} + (t_1 b_2)^{\alpha_1} < \lambda c_1^{\beta_1}$ $n^{\alpha_2} + (t_2 b_1)^{\alpha_2} < \lambda c_2^{\beta_2}$ | det(J) | + | + | - | - | + |
| | | tr(J) | - | + | +/- | +/- | 0 |
| | | Stability | ESS | Unstable point | Saddle point | Saddle point | Disequilibrium point |
| 4 | $m^{\alpha_1} + (t_1 b_2)^{\alpha_1} < \lambda c_1^{\beta_1}$ $n^{\alpha_2} + (t_2 b_1)^{\alpha_2} < \lambda c_2^{\beta_2}$ | det(J) | + | - | - | + | - |
| | | tr(J) | - | +/- | +/- | + | 0 |
| | | Stability | ESS | Saddle point | Saddle point | Unstable point | Saddle point |

Supplier and retailer, as decision-makers, are mainly influenced by factors such as costs of information sharing, collaboration benefits and additional benefits of information sharing. In reality, decision makers are limited rationality when making decisions about information sharing, especially under the influence of various uncertainties. They are risk-averse in the face of gains, and risk-appetizing when faced with losses, and their perception of losses is stronger. The main factors that affect the convergence of the system to the optimal desired strategy (sharing, sharing) are analyzed below.

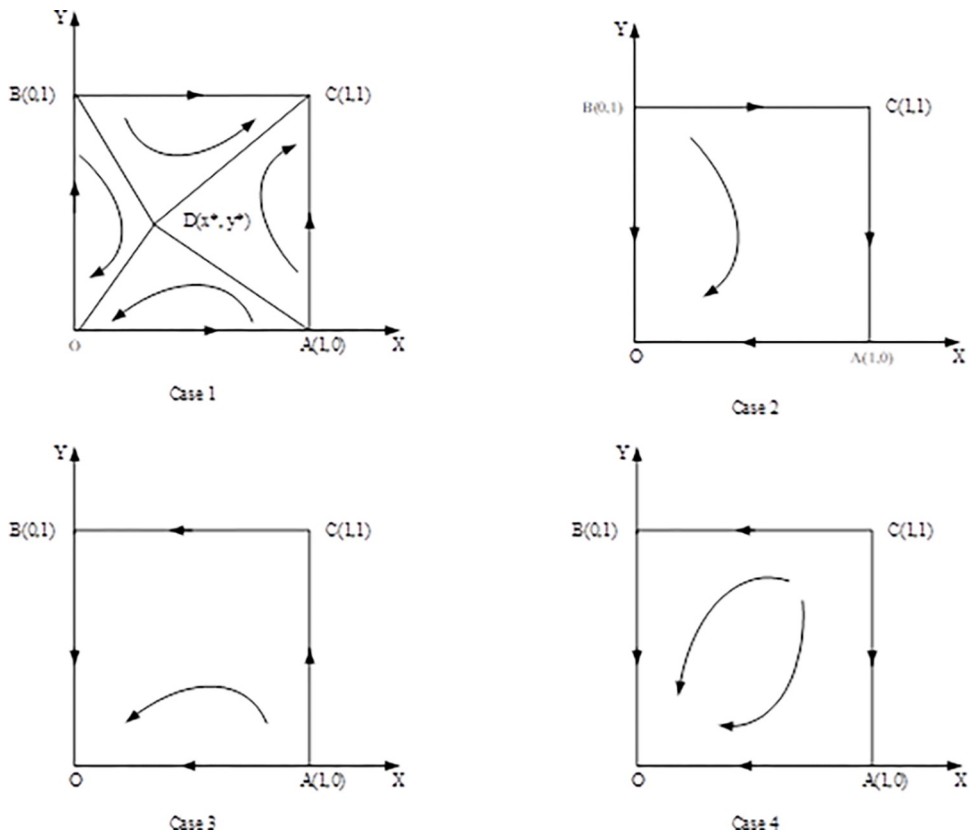

**Fig 1. Phase diagram of the system in different cases.**

**Proposition 1:** When both the supplier and the retailer choose information sharing, the higher the cost of information sharing, the smaller the probability that the system will converge to the stable strategy (sharing, sharing).

**Proof:** Calculating the first-order partial derivatives of $c_1$ and $c_2$ in Formula (11), $\frac{\partial P}{\partial c_1} = -\frac{\lambda \beta_1 c_1^{\beta_1-1}}{2[m^{\alpha_1}+(t_1 b_2)^{\alpha_1}]} < 0$ and $\frac{\partial P}{\partial c_2} = -\frac{\lambda \beta_2 c_2^{\beta_2-1}}{2[n^{\alpha_2}+(t_2 b_1)^{\alpha_2}]} < 0$ can be obtained. Thus, the higher the cost of information sharing, the more unfavorable it is for supplier and retailer to carry out information sharing and cooperation.

**Proposition 2:** The higher the additional income of long-term information sharing for supplier and retailer, the greater the probability that the system will converge to the stable strategy (sharing, sharing).

**Proof:** Calculating the first-order partial derivatives of $m$ and $n$ in Formula (11), $\frac{\partial P}{\partial m} = \frac{\lambda c_1^{\beta_1} (2\alpha_1 m^{\alpha_1}/m)}{[2m^{\alpha_1}+2(t_1 b_2)^{\alpha_1}]^2} > 0$ and $\frac{\partial P}{\partial n} = \frac{\lambda c_2^{\beta_2} (2\alpha_2 n^{\alpha_2}/n)}{[2n^{\alpha_2}+2(t_2 b_1)^{\alpha_2}]^2} > 0$ can be obtained. Thus, if suppliers and retailers can feel the improvement of reputation and increase of market share when they choose information sharing strategies, they will further information sharing.

**Proposition 3:** The collaboration benefits of information sharing between supplier and retailer will be influenced by the complementarity of knowledge and the amount of information sharing. The greater the collaborative benefits obtained by both parties, the greater the probability that the system converges to the stable strategy (sharing, sharing).

**Proof:** $\frac{\partial P}{\partial t_1} = \frac{\lambda c_1^{\beta_1} [2\alpha_1(t_1 b_2)^{\alpha_1}/t_1]}{[2m^{\alpha_1}+2(t_1 b_2)^{\alpha_1}]^2} > 0$, $\frac{\partial P}{\partial t_2} = \frac{\lambda c_2^{\beta_2} [2\alpha_2(t_2 b_1)^{\alpha_2}/t_2]}{[2n^{\alpha_2}+2(t_2 b_1)^{\alpha_2}]^2} > 0$, $\frac{\partial P}{\partial b_1} = \frac{\lambda c_2^{\beta_2} [2\alpha_2(t_2 b_1)^{\alpha_2}/b_1]}{[2n^{\alpha_2}+2(t_2 b_1)^{\alpha_2}]^2} > 0$ and

$\frac{\partial P}{\partial b_2} = \frac{\lambda c_1^{\beta_1} [2\alpha_2(t_1 b_2)^{\alpha_1}/b_2]}{[2n^{\alpha_1}+2(t_1 b_2)^{\alpha_1}]^2} > 0$. Thus, the higher the amount of information shared by supplier and retailer and the degree of complementarity of the shared information, the more favorable it is to promote information sharing between the two decision-making subjects.

**Proposition 4:** The larger the loss sensitivity coefficients $\lambda$ of supplier and retailer, the smaller the probability that the system converges to the stable strategy (sharing, sharing).

**Proof:** $\frac{\partial P}{\partial \lambda} = -\frac{c_2^{\beta_2}}{2[n^{\alpha_2}+(t_2 b_1)^{\alpha_2}]} - \frac{c_1^{\beta_1}}{2[m^{\alpha_1}+(t_1 b_2)^{\alpha_1}]} < 0$. Thus, there is a negative correlation between $P$ and $\lambda$. This shows that the more sensitive supplier and retailer are to losses, the more unfavorable it is for information sharing.

**Proposition 5:** The less risk-averse the supplier and retailer are in the face of revenue (that is, the larger $\alpha_i(i = 1,2)$ is), the greater the probability that the system will converge to the stable strategy (sharing, sharing).

**Proof:** $\frac{\partial P}{\partial \alpha_1} = \frac{\lambda c_1^{\beta_1} [2m^{\alpha_1}\ln(m)+2(t_1 b_2)^{\alpha_1}\ln(t_1 b_2)]}{[2m^{\alpha_1}+2(t_1 b_2)^{\alpha_1}]^2} > 0$ and $\frac{\partial P}{\partial \alpha_2} = \frac{\lambda c_2^{\beta_2} [2n^{\alpha_2}\ln(n)+2(t_2 b_1)^{\alpha_2}\ln(t_2 b_1)]}{[2n^{\alpha_2}+2(t_2 b_1)^{\alpha_2}]^2} > 0$. Thus, there is a positive correlation between $P$ and $\alpha_i$. This shows that the greater the degree of risk appetite (that is, the more rational) supplier and retailer show in the face of benefits, the more conducive to the occurrence of information sharing behavior.

**Proposition 6:** The smaller the risk preference of supplier and retailer in the face of losses (that is, the larger $\beta_i(i = 1,2)$ is), the smaller the probability that the system will converge to the stable strategy (sharing, sharing).

**Proof:** $\frac{\partial P}{\partial \beta_1} = -\frac{\lambda \beta_1 c_1^{\beta_1-1}}{2[m^{\alpha_1}+(t_1 b_2)^{\alpha_1}]} < 0$ and $\frac{\partial P}{\partial \beta_2} = -\frac{\lambda \beta_2 c_2^{\beta_2-1}}{2[n^{\alpha_2}+(t_2 b_1)^{\alpha_2}]} < 0$. Thus, there is a negative correlation between $P$ and $\beta_i$. This shows that the greater the degree of risk appetite (that is, the more irrational) supplier and retailer show in the face of losses, the more conducive to the occurrence of information sharing behavior.

The reason why the information sharing system is difficult to reach the optimal state is analyzed as follows:

1. This study is based on the analysis of the two-stage supply chain. However, there are multiple suppliers and retailers in the actual market. The market is complex and has many uncertainties, and there is information asymmetry among multiple parties, which makes decision-making subjects such as suppliers and retailers in the supply chain have cognitive biases. That is to say, they will underestimate the potential benefits of information sharing (increasing of market share and reputation), and overestimate the maintenance costs of information sharing.

2. Information sharing under supply chain collaboration involves many stakeholders, and the degree of trust among them is low. The subjects know little about each other's ability and moral level in the initial stage of the formation of the supply chain.

3. Based on the limited rationality of decision makers in reality and the utility of the prospect theory, when the information sharing subject faces loss and gain, the negative utility brought by the loss is greater than the positive utility brought by the gain. Moreover, the decision-making subject will overestimate the loss utility of maintenance costs and underestimate the benefit utility of information collaboration to a certain extent.

4. Finally, the uncertainty in the environment in which the supply chain operates may also play a role. Information collaboration benefits depend not only on the amount of information shared and the information absorption and transformation capabilities of both parties but are also influenced by external turbulent environments. However, uncertain environmental factors can have both positive and negative effects, making it difficult to intuitively assess their impact on the system's evolutionary outcomes. Addressing this potential factor is a focus of future research for the author.

## 3. Influencing factors of behavior evolution and simulation analysis

To provide a more intuitive analysis of the impact of factors such as information sharing costs, additional gains, collaborative gains, and loss aversion coefficients on the evolutionary game results of suppliers and retailers under the conditions of prospect theory, we utilized MATLAB software for simulation modeling. MATLAB, recognized as a sophisticated mathematical computing and programming environment, finds extensive application across scientific, engineering, data analysis, and machine learning domains. Its robust mathematical computing capabilities, coupled with an array of rich toolboxes, enable parallel processing. Furthermore, its potent graphical and visualization functions were pivotal in selecting it as the primary software for simulation analysis in this study.

Literatures [11, 32] were referenced for parameter setting. We referred to existing literature for the following reasons: 1. The parameter settings in the literature are founded on established theories, with authors validating the effectiveness of these parameters. 2. The literature we consulted exhibits a degree of relevance to our research, offering valuable insights for parameter configuration. 3. These parameter configurations yield optimal visual analytical outcomes. In addition, the data has been adjusted repeatedly to achieve a good visual analysis effect. Consequently, the loss aversion coefficient is $\lambda = 2.25$, risk coefficients are $\alpha_1 = \alpha_2 = \beta_1 = \beta_2 = 0.88$, amount of information shared by suppliers and retailers are $b_1 = b_2 = 250$, complementarity coefficients are $t_1 = t_2 = 0.5$, information sharing costs are $c_1 = c_2 = 30$, additional benefits are $m = n = 100$, initial information sharing probability of the supplier is $x = 0.5$ and initial information sharing probability of the retailer is $y = 0.5$.

1. The evolutionary behavior path of game subjects
   The impact of different initial probabilities on the final decision of supplier and retailer and

the dynamic replication system is analyzed by simulating their decision-making process under different initial probabilities. It can be seen that when the parameter values satisfy the corresponding constraints, no matter how the initial probability changes between 0 and 1, the game subject will choose the behavior strategy that can maximize the benefit (Fig 2A). In this case, the system will eventually stabilize at an equilibrium state of (non-sharing, non-sharing) and (sharing, sharing). The probability x of the supplier choosing the information sharing strategy is fixed at 0.5, and the corresponding probability y of the retailer is fixed at 0.5 to explore the impact of the initial probability change on their decision-making. It can be observed that the higher initial probabilities for suppliers and retailers expedite the rate of convergence toward cooperation (Fig 2B and 2C). This outcome underscores that when the willingness of both parties for initial information sharing is low, it hampers inter-enterprise information sharing. Hence, whether acting as a supplier or retailer, proactive engagement in constructive information exchange with supply chain counterparts, the establishment of trust mechanisms, and the actualization of information sharing are advisable.

2. The impact of the change in the information sharing cost $c_1$ on the evolution result
The gradual increase of the information sharing cost $c_1$, the result of $x$ converging to 1 gradually changes to converging to 0. In addition, there is a critical value of information sharing cost, which is between 40 and 50. When the information sharing cost is less than this critical value, $x$ converges to 1. The smaller the value of $c_i$, the faster the convergence speed (Fig 3). Hence, elevated information sharing costs serve as a deterrent to inter-enterprise information sharing. As costs rise, enterprises become less inclined to engage in information sharing. At this juncture, it is imperative to harness advanced technologies such as 5G, artificial intelligence, and the Internet of Things (IoT) to reduce costs. This, in turn, ensures mutual information sharing and fosters supply chain collaboration. In summary, there exists an inverse correlation between the probability of system convergence to sharing and information sharing costs. Lowering information sharing costs and enhancing efficiency can encourage proactive information sharing among decision-makers.

3. The impact of the change in the additional benefit $m$ on the evolution result
The gradual increase of the additional benefit $m$, the result of $x$ and $y$ converging to 0 gradually changes to converging to 1. In addition, there is a critical value of additional benefit, which is between 20 and 40. When the additional benefit is less than this critical value, $x$ converges to 0. When the additional benefit is greater than this critical value, $x$ converges to 1. The smaller the value of $m$, the faster the convergence speed (Fig 4). This outcome signifies that when supply chain enterprises anticipate higher additional gains from information sharing, they are more inclined to partake in it. Information sharing within the supply chain grants retailers access to precise market demand information, given their proximity to the market, while suppliers hold inventory, transportation, and logistics data. The exchange of information between both parties further augments market share, thereby increasing additional gains. Consequently, supply chain enterprises should promptly share information conducive to enhancing market share, including market demand forecasts, inventory information, transportation, and logistics data, among others. This sharing, however, should exclude information pertaining to each entity's core competitive advantages.

4. The impact of changes in collaborative benefits on the evolution result
The collaboration benefit is determined by the amount of information shared by both parties $b_1$ and $b_2$, and the coefficient of complementarity $t_1$ and $t_2$. According to the above analysis, the amount of information sharing and the coefficient of complementarity have the

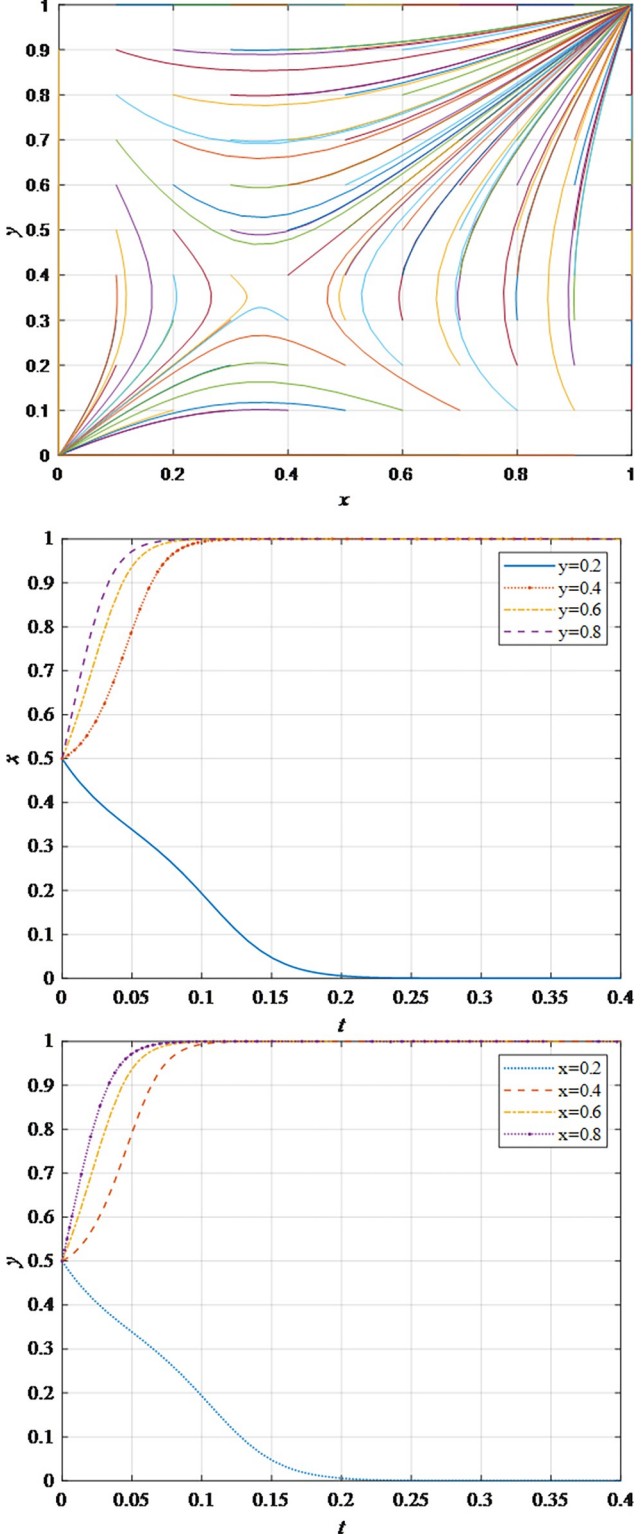

**Fig 2.** a. Effect of different initial probabilities on the stability of the system. b. The impact of retailer's initial probability change on supplier's strategy selection. c. The impact of supplier's initial probability change on retailer's strategy selection.

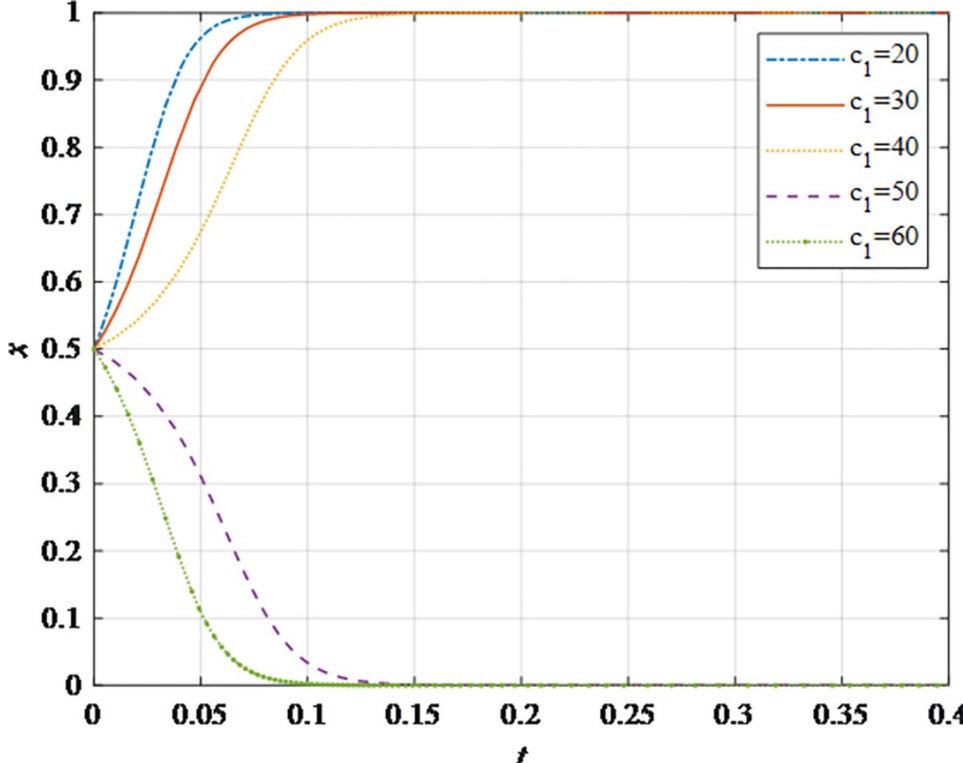

**Fig 3. The impact of information maintenance cost $c_1$ on the evolution result.**

same effect on the trend of evolution results in terms of collaboration benefits. In this study, the supplier is taken as the object for analysis, the complementarity coefficient $t_1$ is fixed at 0.5 and the amount of information sharing $b_2$ is changed for a more intuitive observation. It can be seen from Fig 5 that with the gradual increase of the amount of information sharing, the result of $x$ converging to 0 gradually changes to converging to 1. In addition, there is a critical value between 100 and 150. When the value is less than this critical value, $x$ converges to 0. When the value is greater than this critical value, $x$ converges to 1. The larger the amount of the information sharing, the faster the convergence speed (Fig 5). Thus, there is a positive correlation between the probability that the system converges to sharing and the amount of information sharing. This underscores that for supply chain enterprises aiming to maximize profits and market share, increasing the quantity and frequency of information sharing is imperative. This information should encompass market demand forecasts, inventory data, transportation, logistics information, and other insights conducive to optimizing decision-making. However, it should exclude details pertaining to each entity's core competitive strengths.

5. The impact of the change in the loss sensitivity coefficient $\lambda$ on the evolution result
According to the prospect theory, most people are more sensitive to loss than income, and the sensitivity coefficient is greater than 1. The gradual increase of $\lambda$, the result of $x$ converging to 1 gradually changes to converging to 0. There is a critical value between 3 and 3.5. When the loss sensitivity coefficient is less than the critical value, $x$ converges to 1. The smaller the value, the faster its convergence speed (Fig 6). It can be seen from the results that the smaller the loss sensitivity coefficient of the information sharing subject, the easier it is to share information and achieve information collaboration. The results emphasize that

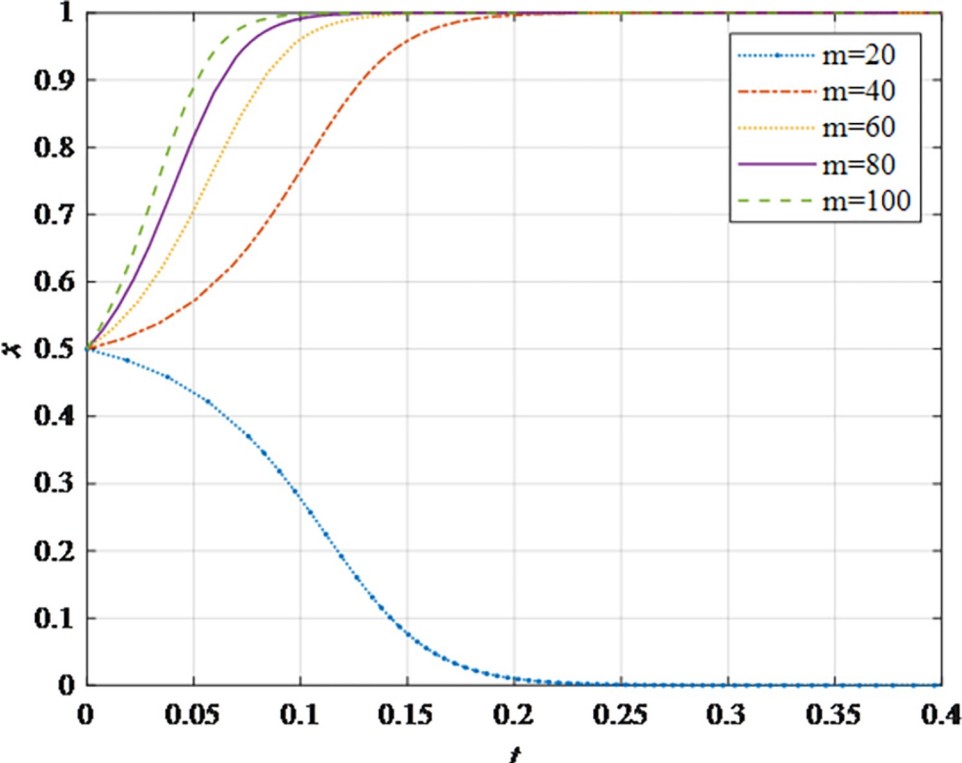

**Fig 4. The impact of the additional benefit $m$ on the evolution result.**

when supply chain enterprises engage in information sharing, they should strive to mitigate their sensitivity to losses. The focus should remain on long-term gains and development, without undue concern for short-term cost investments. Attention to long-term returns and growth should take precedence over heightened sensitivity to initial costs.

6. The impact of the change in risk aversion coefficient facing benefits $\alpha_1$ on the evolution result

   According to the prospect theory, most people tend to avoid risks in the face of benefits. The smaller the value of $\alpha_1$, the greater the degree of risk aversion. The gradual increase of $\alpha_1$, the result of $x$ converging to 0 gradually changes to converging to 1. In addition, there is a critical value between 0.8 and 0.88. When the value is less than this critical value, $x$ converges to 0. When the value is greater than this critical value, $x$ converges to 1. The larger the value of $\alpha_1$, the faster the convergence speed (Fig 7). Therefore, the more rational the information sharing parties show in the face of benefits, the easier it is to choose "information sharing". The findings suggest that when supply chain enterprises engage in information sharing and stand to gain from it, they should not solely focus on short-term gains. Instead, they should prioritize long-term gains and development. In situations involving potential gains, they should further invest effectively to maximize returns.

7. The impact of the change in risk aversion coefficient facing losses $\beta_1$ on the evolution result

   According to the prospect theory, most people are risk appetite in the face of losses. The smaller the value of $\beta_1$, the more inclined to risk preference. The gradual increase of the value, the result of $x$ converging to 1 gradually changes to converging to 0. In addition, there is a critical value between 0.88 and 0.95. When the value is less than this critical value, $x$ converges to 1. When the value is greater than this critical value, $x$ converges to 0. The

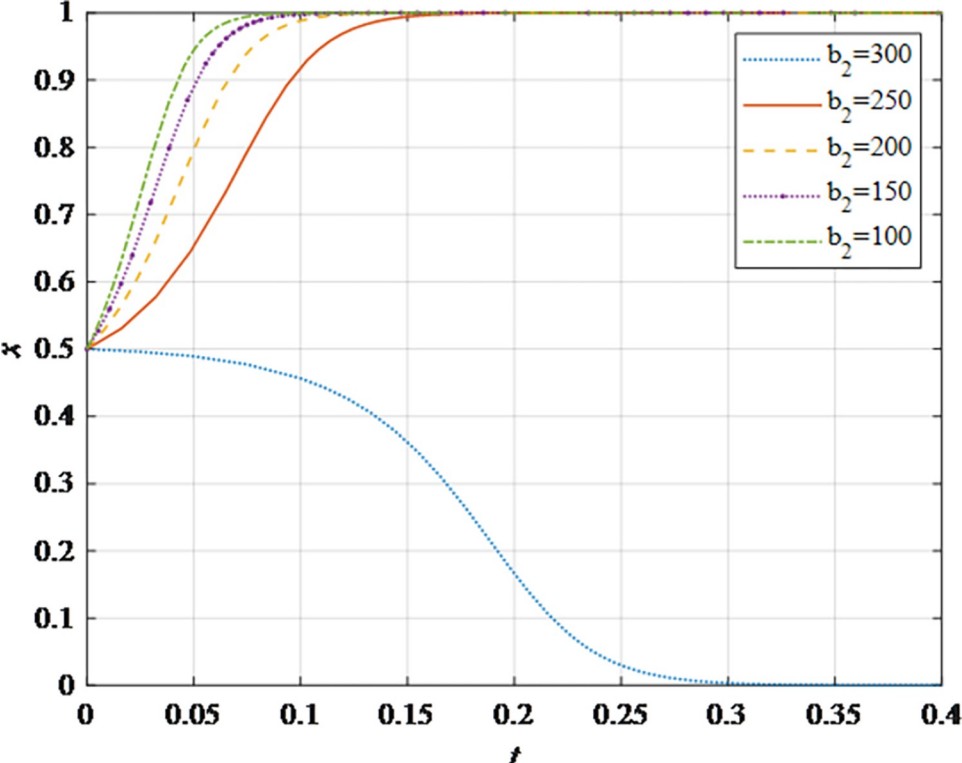

**Fig 5. The impact of the amount of information sharing $b_2$ on the evolution result.**

larger the value of $\beta_1$, the faster the convergence speed (Fig 8). These findings underscore that, when supply chain enterprises engage in information sharing that might result in losses, they should strive to temper their emotional response to such losses. While risk aversion can stimulate the occurrence of information-sharing behaviors, it may inadvertently lead to the neglect of larger potential losses stemming from unforeseen risks. To address this, the implementation of risk-sharing mechanisms can serve as a more effective means of risk management.

## 4. Conclusion and management implications

This study reveals that previous research has predominantly focused on constructing payoff matrices based on objective gains, overlooking the influence of psychological perceptions of supply chain collaborative decision-making entities on their choices [11, 33, 34]. Supply chain decision-making entities' judgments of the probability of future events, benefits, or losses and their attitudes towards risk will differ due to the complex decision-making environment and uncertainty, and subjective psychological perceptions will affect strategies. Hence, this paper combines the prospect theory and game theory under the bounded rationality hypothesis with the relevant research foundation of predecessors, constructs game theory model based on information sharing behavior of secondary supply chain entities, and conducts simulation research. The following conclusions and implications were obtained:

### 4.1 Research conclusion

The study found that the decision-making of information sharing behavior among supply chain collaborative entities under information sharing costs, collaborative benefits, and the

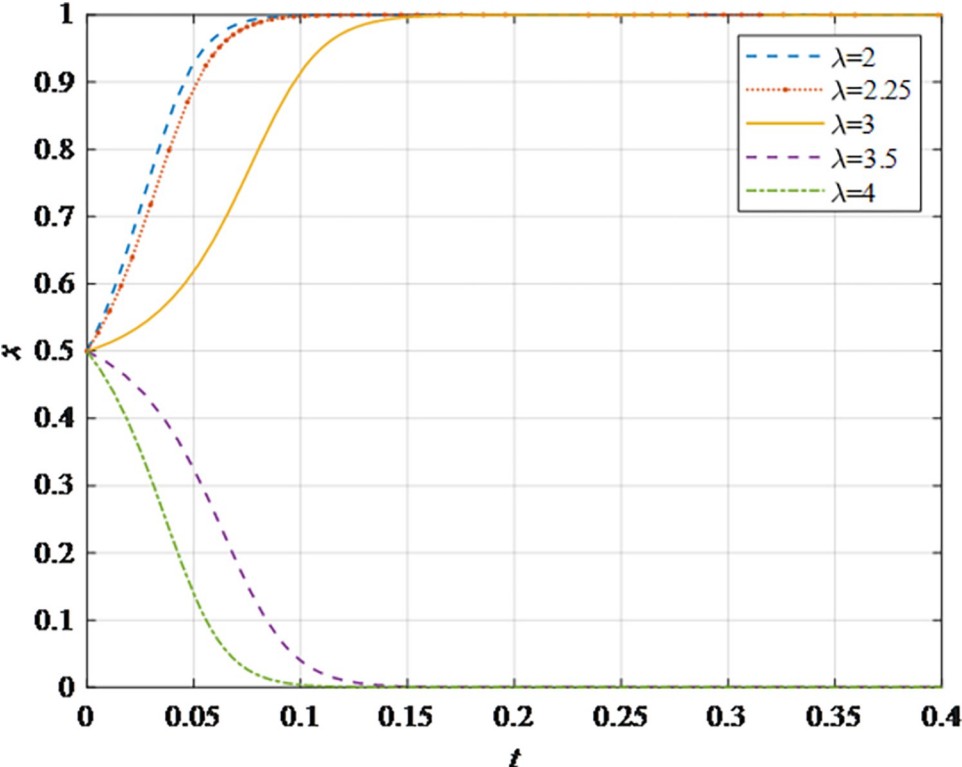

**Fig 6. The impact of the loss sensitivity coefficient $\lambda$ on the evolution result.**

psychological perception of decision-makers facing benefits and losses are influenced by various factors. The probability of the decision-making process under information sharing behavior of supply chain collaborative entities is negatively related to information sharing costs, and positively related to the collaborative benefits generated by information sharing. At the same time, under information sharing in supply chain collaboration, the smaller the degree of risk aversion facing the benefit risk of supply chain decision-making entities, the more rational they are and the more favorable it is for information sharing behavior to occur. The greater the risk preference facing loss, the more irrational they are, and the more favorable it is for information sharing behavior to occur. The smaller the sensitivity of the supply chain decision-making entity to losses, the more favorable it is for the system to evolve to the state of information sharing.

### 4.2 Management implications

This paper proposes the following management implications:

1. Information sharing among supply chain entities is influenced by a multitude of factors, including members' knowledge acquisition capabilities, partner relationships, digitalization levels, and information heterogeneity. To enhance the benefits derived from inter-enterprise information sharing, there should be a strengthening of inter-firm collaborations and a concerted effort to enhance the learning and information integration abilities of supply chain members.

2. Recognizing the pivotal role of information sharing in supply chain collaboration, management should foster an open information flow between suppliers and retailers to bolster cooperation and overall supply chain efficiency.

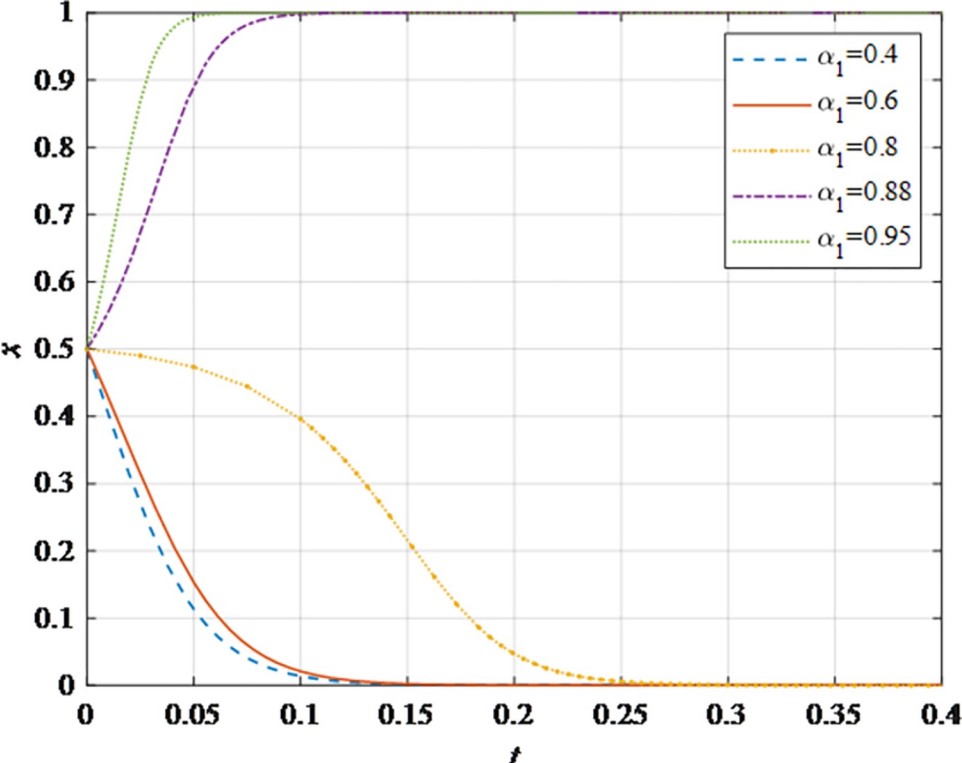

**Fig 7. The impact of the risk aversion coefficient $\alpha_1$ on the evolution result.**

3. High information maintenance costs can impede information sharing within collaborative supply chains. To mitigate these costs, such as communication overheads, advanced technologies like 5G, artificial intelligence, and the Internet of Things (IoT) can be employed to establish collaborative information sharing platforms. This, in turn, can stimulate a proactive approach to information exchange and cultivate a spirit of mutual cooperation.

4. Notably, the research underscores differences in organizational behavior psychology regarding losses and gains among members. Therefore, it is advisable to engage professionals versed in organizational behavior psychology and risk analysis to provide training and elevate employees' psychological resilience. This, in turn, would reduce their sensitivity to losses.

5. In the pursuit of effective supply chain collaboration, a keen focus should be placed on the information integration and collection capabilities of potential partners during the partner selection phase. Improved communication, facilitated by efficient information sharing, can be instrumental in resolving challenges encountered during collaboration.

6. Effective information sharing strategies hinge on trust among supply chain partners. Management efforts should concentrate on fostering trust through shared success stories, transparent contracts, and commitments.

Therefore, to achieve the optimal state of supply chain collaborative information sharing, it is necessary to consider and address the aforementioned factors and challenges, improve the level of trust and understanding among all entities, rectify decision makers' cognitive biases, and adapt to the dynamic and uncertain market environment.

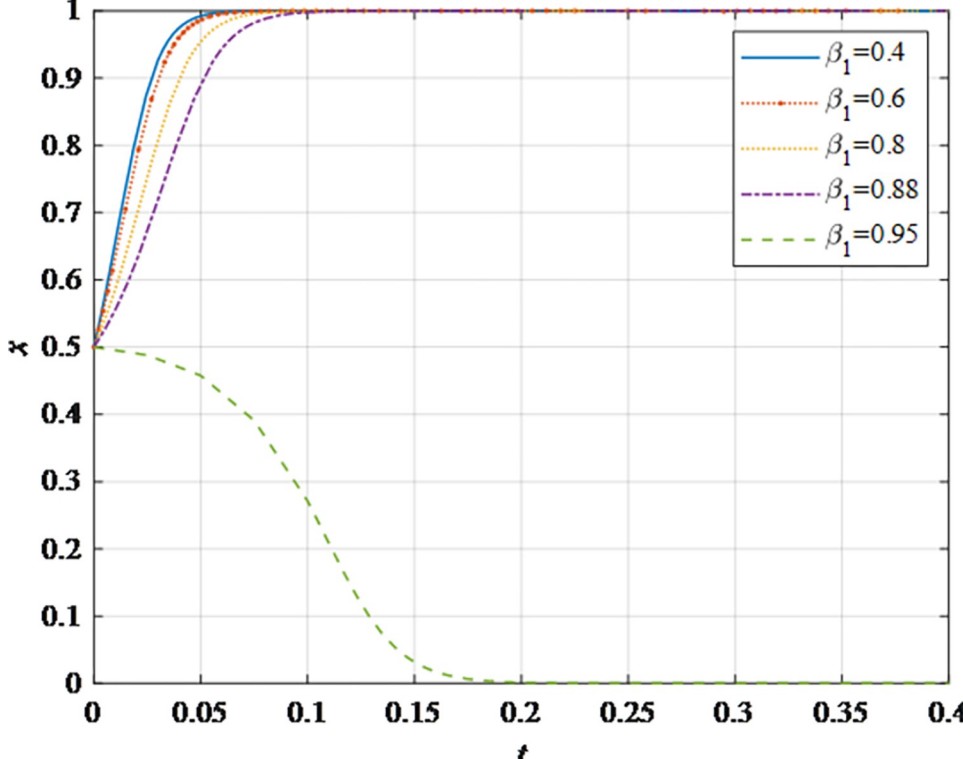

**Fig 8. The impact of the risk aversion coefficient $\beta_1$ on the evolution result.**

### 4.3 Prospects for future research

This study has solely addressed the supply chain partnership between a single supplier and a single retailer. In practice, scenarios involving multiple suppliers and retailers are prevalent, each entailing intricate dynamics of interests. Among these, the behavior of information sharing warrants deeper exploration. While this study assumed the independence of various factors influencing information-sharing behaviors among supply chain entities, it's crucial to further investigate potential interconnections or constraints between these factors. Moreover, these factors may exert differing degrees of influence on information-sharing decisions, necessitating future research to encompass this aspect as well.

Furthermore, the supply chain environment is inherently uncertain, with environmental factors capable of both positive and negative impacts. These uncertainties can significantly affect system evolution outcomes, although discerning their precise influence can prove challenging. Future research endeavors may delve into the realm of uncertainty theory and methods to shed light on environmental uncertainties.

## Supporting information

**S1 File.**
(DOCX)

## Author Contributions

**Conceptualization:** Hongcheng Gan.

**Investigation:** Luyu Zhai.

**Project administration:** Hongcheng Gan.

**Visualization:** Luyu Zhai.

**Writing – original draft:** Meng Liu.

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
