## [Decision Letter · Decision Letter 0]

6 Sep 2023

PONE-D-23-26347Analysis of the Evolutionary Process of Multi-agent Information Sharing Behavior under Supply Chain CollaborationPLOS ONE

Dear Dr. Gan,

Thank you for submitting your manuscript to PLOS ONE. After careful consideration, we feel that it has merit but does not fully meet PLOS ONE’s publication criteria as it currently stands. Therefore, we invite you to submit a revised version of the manuscript that addresses the points raised during the review process.

Dear authors,

Thank you for choosing PLOS ONE as an outlet for your paper publication. We have received evaluations from experts. The reviewers have highlighted several issues which need to be answered and strengthen the quality of this draft.

Authors should take these comments into account and adjust the manuscript content accordingly.

1. The draft needs to be significantly improved. In addition to the improvements requested by Reviewers, you should expand them in line with the PLOS ONE guidelines and revise the abstract accordingly.

2. Reviewers highlight several issues concerning the introduction, literature review, and conclusion sections. Therefore, Revising these sections to meet the publication criteria of PLOS ONE is suggested.

3. This article requires proofreading to enhance the linguistic quality further. Also, please update the reference list, preferably referencing recent works published in top leading journals.

4. I suggest authors ensure that all the cited articles should be properly listed in the reference section.

We look forward to receiving your revised manuscript.

Kind regards,

Syed Abdul Rehman Khan, PhD

Academic Editor

PLOS ONE

Journal Requirements:

1. When submitting your revision, we need you to address these additional requirements. Please ensure that your manuscript meets PLOS ONE's style requirements, including those for file naming. The PLOS ONE style templates can be found at https://journals.plos.org/plosone/s/file?id=wjVg/PLOSOne_formatting_sample_main_body.pdf and https://journals.plos.org/plosone/s/file?id=ba62/PLOSOne_formatting_sample_title_authors_affiliations.pdf 2. Please note that PLOS ONE has specific guidelines on code sharing for submissions in which author-generated code underpins the findings in the manuscript. In these cases, all author-generated code must be made available without restrictions upon publication of the work. Please review our guidelines at https://journals.plos.org/plosone/s/materials-and-software-sharing#loc-sharing-code and ensure that your code is shared in a way that follows best practice and facilitates reproducibility and reuse. 3. We note that the grant information you provided in the ‘Funding Information’ and ‘Financial Disclosure’ sections do not match.  When you resubmit, please ensure that you provide the correct grant numbers for the awards you received for your study in the ‘Funding Information’ section.
 4. In your Data Availability statement, you have not specified where the minimal data set underlying the results described in your manuscript can be found. PLOS defines a study's minimal data set as the underlying data used to reach the conclusions drawn in the manuscript and any additional data required to replicate the reported study findings in their entirety. All PLOS journals require that the minimal data set be made fully available. For more information about our data policy, please see http://journals.plos.org/plosone/s/data-availability. Upon re-submitting your revised manuscript, please upload your study’s minimal underlying data set as either Supporting Information files or to a stable, public repository and include the relevant URLs, DOIs, or accession numbers within your revised cover letter. For a list of acceptable repositories, please see http://journals.plos.org/plosone/s/data-availability#loc-recommended-repositories. Any potentially identifying patient information must be fully anonymized. Important: If there are ethical or legal restrictions to sharing your data publicly, please explain these restrictions in detail. Please see our guidelines for more information on what we consider unacceptable restrictions to publicly sharing data: http://journals.plos.org/plosone/s/data-availability#loc-unacceptable-data-access-restrictions. Note that it is not acceptable for the authors to be the sole named individuals responsible for ensuring data access. We will update your Data Availability statement to reflect the information you provide in your cover letter. 5. PLOS requires an ORCID iD for the corresponding author in Editorial Manager on papers submitted after December 6th, 2016. Please ensure that you have an ORCID iD and that it is validated in Editorial Manager. To do this, go to ‘Update my Information’ (in the upper left-hand corner of the main menu), and click on the Fetch/Validate link next to the ORCID field. This will take you to the ORCID site and allow you to create a new iD or authenticate a pre-existing iD in Editorial Manager. Please see the following video for instructions on linking an ORCID iD to your Editorial Manager account: https://www.youtube.com/watch?v=_xcclfuvtxQ

Reviewers' comments:

Reviewer's Responses to Questions

**Comments to the Author**

1. Is the manuscript technically sound, and do the data support the conclusions?

Reviewer #1: Yes

Reviewer #2: Yes

2. Has the statistical analysis been performed appropriately and rigorously? 

Reviewer #1: Yes

Reviewer #2: Yes

3. Have the authors made all data underlying the findings in their manuscript fully available?

Reviewer #1: Yes

Reviewer #2: Yes

4. Is the manuscript presented in an intelligible fashion and written in standard English?

Reviewer #1: No

Reviewer #2: Yes

5. Review Comments to the Author

Reviewer #1: This study conducted on the Analysis of the Evolutionary Process of Multi-agent Information Sharing Behavior under Supply Chain Collaboration. I found the article required significant improvement in revision phase. Following are my suggestions:

1. Title of this article is fine but it would be better to make it more attractive.

2. Abstract needs to be revise and further include the key findings of this study.

3. Introduction section is not properly covering all the parameters of this study. Therefore, I recommend to include clearly "Research objective" . it will be better to include a subheading of "Research objective" in introduction section and explain the research objective.

4. Authors should include and cite the up-to-date relevant articles published in top ranked journals. Also, it would be better to include the published articles, which adopted same methodology in the domain of supply chain. Following articles can be considered:

https://doi.org/10.1016/j.procs.2022.09.461

https://doi.org/10.1080/13675567.2021.1879752

https://doi.org/10.1016/j.jenvman.2023.117968

5. Managerial implications are not sufficient. Please extend it and further include new implications.

6. This article required English proofread.

Reviewer #2: This study aims to explore the dynamic evolution process and influencing factors of multi-agent information sharing behavior decision-making under supply chain collaboration. This paper constructed an evolutionary game model with suppliers and retailers as the main subjects, using a combination of game theory and prospect theory, and constructed an information sharing prospect value matrix by combining prospect value functions and weight functions. The study is interesting. However, there are some issues that need to be revised. My specific comments are as follows:

1. Overall, the writing could be further improved. There are some grammatical errors/typos to be aware of. Please revise them carefully. e.g.

“However, in this study, it is believed that its value depend on …”

Inconsistent formatting of formula numbering.

2. The motivation and contribution of this study should be clearly highlighted in the introduction. The current statement in this portion is too weak.

3. As stated by the authors, this study distinguishes from existing research by the subjects' behavioral decisions on information sharing from the perspective of supply chain cooperation. Then, the authors should explain in detail in the manuscript how this study embodies supply chain collaboration in addition to information sharing and what collaboration factors are considered.

4. The picture of fig. 1 should be on the same page as the figure notes.

5. The authors should describe the compilation environment used for the simulation experiment.

6. The authors should explain why the parameter settings in literature [14,38] are used.

7. How the uncertainty of the environment is represented in the study.

8. The authors should analyze the results of the experiment to help readers understand the meaning and results of the experiment, rather than just describing the content of the picture.

6. PLOS authors have the option to publish the peer review history of their article (what does this mean?). If published, this will include your full peer review and any attached files.

Reviewer #1: No

Reviewer #2: No

---

## [Author Response · Author response to Decision Letter 0]

16 Oct 2023

Response to Reviewers

Dear Syed Abdul Rehman Khan, PhD

Title：An Evolutionary Analysis of Supply Chain Collaborative Information Sharing Based on Prospect Theory

Authors：Meng Liu, Luyu Zhai, Hongcheng Gan

Dear Syed Abdul Rehman Khan, PhD

We would like to thank you for the prompt review of our paper and for the opportunity to respond to Academic Editor, Reviewers #1, #2. All reviewers provided us with insightful and valuable comments. We have revised our paper and addressed all of the issues raised by all reviewers, based on their comments and suggestions. We have attached a point by point response for the reviewers. Please see the details bellow. Thank you again for giving us the opportunity to revise the paper. Suggestions and comments from the reviewers have further significantly improved our paper.

Sincerely,

The Authors

Encl.

(1) Responses to comments of Academic Editor

(2) Responses to comments of Reviewer #1

(3) Responses to comments of Reviewer #2

Responses to Comments of Academic Editor

We would like to thank you for your time spent in reviewing our paper and for providing us with valuable comments and suggestions. We also appreciate your positive view on our results and hope that this version has fully addressed your concerns. To make it easy to follow, we first show (in italic) your comment and then present our response (in normal font).

Comments

1. The draft needs to be significantly improved. In addition to the improvements requested by Reviewers, you should expand them in line with the PLOS ONE guidelines and revise the abstract accordingly.

2. Reviewers highlight several issues concerning the introduction, literature review, and conclusion sections. Therefore, Revising these sections to meet the publication criteria of PLOS ONE is suggested.

3. This article requires proofreading to enhance the linguistic quality further. Also, please update the reference list, preferably referencing recent works published in top leading journals.

4. I suggest authors ensure that all the cited articles should be properly listed in the reference section.

Response/Revision: Firstly，We have made revisions as per the reviewer's request to the abstract, introduction, literature review, and conclusion sections, and have improved the language quality of the article. Additionally, we have updated the references, adding recent publications from top leading journals and removing some references to ensure that all referenced articles are correctly listed in the reference section. We have made modifications to meets PLOS ONE's style requirements, including those for file naming. We have removed incorrect funding numbers and have decided not to include related funding information. Since our data involves assigning values to certain parameters, and these values are already provided in the article, they were referenced from data in previously published literature, and we have explained the reasons for referencing these publications. The corresponding authors have also registered their ORCID iD, which has been validated in Editorial Manager.

Thank you again for the suggestions and comments, which have further improved our paper.

Responses to Comments of Reviewer #1

We would like to thank you for your time spent in reviewing our paper and for providing us with valuable comments and suggestions. We also appreciate your positive view on our results and hope that this version has fully addressed your concerns. To make it easy to follow, we first show (in italic) your comment and then present our response (in normal font).

Comments

Comment #1 1. Title of this article is fine but it would be better to make it more attractive.

Response/Revision: Thanks for the comment, and it is a very correct suggestion.In this revision, I've changed the title of the article to "An Evolutionary Analysis of Supply Chain Collaborative Information Sharing Based on Prospect Theory".

Comment #2 Abstract needs to be revise and further include the key findings of this study.

Response/Revision: Thanks for pointing out! Following your suggestion, Thank you for your suggestions. I have already revised the abstract section of the article and have further include the key findings of this study. Please see the details bellow.

“Abstract

In order to delve into the dynamic evolution process and influencing factors of information sharing decisions among stakeholders under supply chain collaboration, this study constructs an evolutionary game model with suppliers and retailers as the primary entities. Within this model, a combined approach of game theory and prospect theory is employed, integrating prospect value functions and weight functions to create an information sharing prospect value matrix. A comprehensive analysis is conducted on the strategic choices and benefits of entities considering the psychological perception of information sharing, and critical factors influencing the stability of information sharing evolution results are explored through numerical simulations using Matlab. The key findings of this study are as follows: Firstly, from the perspective of supply chain collaboration, the probability of entities evolving into information sharing is negatively correlated with the cost of information sharing and positively correlated with the benefits generated by information coordination. Secondly, looking at supply chain collaboration, entities are more likely to engage in information sharing behavior when they exhibit a lower level of risk aversion, indicating greater rationality, when facing profits; conversely, they are more likely to participate in information sharing when they display a higher degree of risk preference, indicating less rationality, in the face of losses. Furthermore, the lesser sensitivity of suppliers and retailers to losses is more likely to drive the system towards an information-sharing state. Based on the primary findings mentioned above, this study offers recommendations for enhancing trust, constructing information exchange platforms, and adjusting psychological awareness. These suggestions contribute to improving information sharing among entities within the supply chain, thus enhancing the overall efficiency and collaboration of the supply chain.”

Comment #3 Introduction section is not properly covering all the parameters of this study. Therefore, I recommend to include clearly "Research objective" . it will be better to include a subheading of "Research objective" in introduction section and explain the research objective.

Response: Thanks for your comments! In the introduction section of the article, I have added relevant descriptions of the research objectives and contributions in the introduction, and provided further explanations of the research objectives and contributions (In lines 118-136). Please see the details bellow.

“In light of the aforementioned limitations, this study, grounded in the perspective of supply chain collaboration, seeks to uncover the latent psychological and behavioral factors in supply chain information sharing decision-making. Specifically, drawing from the assumption of bounded rationality in game theory[32] and combining it with prospect theory, the study examines how the perception of value and risk aversion by game players influence the patterns of system evolution and stability. Through these investigations, we aim to provide a deeper understanding for enterprises and supply chain practitioners, enabling them to refine their decision-making and practices. Furthermore, we hope this research contributes to enriching the foundational theoretical framework of supply chain collaboration and offers new perspectives and scientific methods to researchers in the field.

The main contributions of this study are as follows: 1. It analyzes the behavioral decisions of supply chain decision-makers from the perspective of supply chain collaboration, offering a novel viewpoint. 2. By introducing prospect theory into the domain of supply chain collaboration management, the study provides a fresh theoretical framework that elucidates the evolutionary process of information sharing in supply chains, enriching the existing supply chain theory. 3. This research can assist supply chain decision-makers in better understanding the impact of psychological factors on their decision-making, providing effective guidance for practice. 4. The study also presents research insights and future prospects, serving as a reference and offering suggestions for related researchers.”

Comment #4 Authors should include and cite the up-to-date relevant articles published in top ranked journals. Also, it would be better to include the published articles, which adopted same methodology in the domain of supply chain. Following articles can be considered:

https://doi.org/10.1016/j.procs.2022.09.461

https://doi.org/10.1080/13675567.2021.1879752

https://doi.org/10.1016/j.jenvman.2023.117968

Response/Revision: Thanks for comments! We have cited the up-to-date relevant articles published in top ranked journals.And citing the articles you recommended, the relevant articles have already been indicated in the references section.

Comment #5 Managerial implications are not sufficient. Please extend it and further include new implications.

Response/Revision: Thanks for the comment! This is a very good suggestion. We have revised the relevant managerial implications and expanded to further include new implications (In lines 500-529). Please see the details bellow.

“4.2 Management implications 

This paper proposes the following management implications: 

(1)Information sharing among supply chain entities is influenced by a multitude of factors, including members' knowledge acquisition capabilities, partner relationships, digitalization levels, and information heterogeneity. To enhance the benefits derived from inter-enterprise information sharing, there should be a strengthening of inter-firm collaborations and a concerted effort to enhance the learning and information integration abilities of supply chain members.

(2) Recognizing the pivotal role of information sharing in supply chain collaboration, management should foster an open information flow between suppliers and retailers to bolster cooperation and overall supply chain efficiency.

(3) High information maintenance costs can impede information sharing within collaborative supply chains. To mitigate these costs, such as communication overheads, advanced technologies like 5G, artificial intelligence, and the Internet of Things (IoT) can be employed to establish collaborative information sharing platforms. This, in turn, can stimulate a proactive approach to information exchange and cultivate a spirit of mutual cooperation.

(4) Notably, the research underscores differences in organizational behavior psychology regarding losses and gains among members. Therefore, it is advisable to engage professionals versed in organizational behavior psychology and risk analysis to provide training and elevate employees' psychological resilience. This, in turn, would reduce their sensitivity to losses.

(5) In the pursuit of effective supply chain collaboration, a keen focus should be placed on the information integration and collection capabilities of potential partners during the partner selection phase. Improved communication, facilitated by efficient information sharing, can be instrumental in resolving challenges encountered during collaboration.

(6) Effective information sharing strategies hinge on trust among supply chain partners. Management efforts should concentrate on fostering trust through shared success stories, transparent contracts, and commitments.

Therefore, to achieve the optimal state of supply chain collaborative information sharing, it is necessary to consider and address the aforementioned factors and challenges, improve the level of trust and understanding among all entities, rectify decision makers' cognitive biases, and adapt to the dynamic and uncertain market environment.”

Comment #6 This article required English proofread.

Response/Revision: Thanks for the comment! This is a very good suggestion. We have thoroughly proofread the entire English text.

Thank you again for the suggestions and comments, which have further improved our paper.

Responses to Comments of Reviewer #2

We would like to thank you for your time spent in reviewing our paper and for providing us with valuable comments and suggestions. We also appreciate your positive view on our results and hope that this version has fully addressed your concerns. To make it easy to follow, we first show (in italic) your comment and then present our response (in normal font).

Comments

Comment #1 Overall, the writing could be further improved. There are some grammatical errors/typos to be aware of. Please revise them carefully. e.g.

“However, in this study, it is believed that its value depend on …”

Inconsistent formatting of formula numbering.

Response/Revision: Thanks for the comment! This is a very good suggestion. We have thoroughly proofread the entire English text. We have already changed the formula numbering

Comment #2 The motivation and contribution of this study should be clearly highlighted in the introduction. The current statement in this portion is too weak.

Response: Thanks for your comments! In the introduction section of the article, I have added relevant descriptions of the research motivation and contributions in the introduction, and provided further explanations of the research motivation and contributions (In lines 118-136). Please see the details bellow.

“In light of the aforementioned limitations, this study, grounded in the perspective of supply chain collaboration, seeks to uncover the latent psychological and behavioral factors in supply chain information sharing decision-making. Specifically, drawing from the assumption of bounded rationality in game theory[32] and combining it with prospect theory, the study examines how the perception of value and risk aversion by game players influence the patterns of system evolution and stability. Through these investigations, we aim to provide a deeper understanding for enterprises and supply chain practitioners, enabling them to refine their decision-making and practices. Furthermore, we hope this research contributes to enriching the foundational theoretical framework of supply chain collaboration and offers new perspectives and scientific methods to researchers in the field.

The main contributions of this study are as follows: 1. It analyzes the behavioral decisions of supply chain decision-makers from the perspective of supply chain collaboration, offering a novel viewpoint. 2. By introducing prospect theory into the domain of supply chain collaboration management, the study provides a fresh theoretical framework that elucidates the evolutionary process of information sharing in supply chains, enriching the existing supply chain theory. 3. This research can assist supply chain decision-makers in better understanding the impact of psychological factors on their decision-making, providing effective guidance for practice. 4. The study also presents research insights and future prospects, serving as a reference and offering suggestions for related researchers.”

Comment #3 As stated by the authors, this study distinguishes from existing research by the subjects' behavioral decisions on information sharing from the perspective of supply chain cooperation. Then, the authors should explain in detail in the manuscript how this study embodies supply chain collaboration in addition to information sharing and what collaboration factors are considered.

Response/Revision: We are grateful for your approval of this paper. I explain this issue as follows. In previous research, some scholars have also contended that trust mechanisms, collaborative decision-making, and supply chain visibility are vital determinants of supply chain collaboration. Nevertheless, information sharing has often been considered as the fundamental prerequisite for supply chain collaboration in numerous studies, sometimes even synonymous with collaboration itself. Consequently, in the interest of model simplification, this study exclusively investigates scenarios where supply chain collaboration is achieved through information sharing alone. Your comments have provided the author with valuable insights. Considering other collaborative factors and the interaction between factors is part of the author's future research plans.And we have provided corresponding explanations in the text (In lines 99-105). Please see the details bellow.

“In previous research, some scholars have also contended that trust mechanisms, collaborative decision-making, and supply chain visibility are vital determinants of supply chain collaboration. Nevertheless, information sharing has often been considered as the fundamental prerequisite for supply chain collaboration in numerous studies, sometimes even synonymous with collaboration itself. Consequently, in the interest of model simplification, this study exclusively investigates scenarios where supply chain collaboration is achieved through information sharing alone. ”

Comment #4 The picture of fig. 1 should be on the same page as the figure notes.

Response/Revision: Thanks! I have already restructured the images.

Fig 1. Phase diagram of the system in different cases

Comment #5 The authors should describe the compilation environment used for the simulation experiment.

Response/Revision: Thanks for comments! We have provided a detailed description of the compilation environment used for the simulation experiment (In lines 344-349). Please see the details bellow.

“MATLAB, recognized as a sophisticated mathematical computing and programming environment, finds extensive application across scientific, engineering, data analysis, and machine learning domains. Its robust mathematical computing capabilities, coupled with an array of rich toolboxes, enable parallel processing. Furthermore, its potent graphical and visualization functions were pivotal in selecting it as the primary software for simulation analysis in this study.”

Comment #6 The authors should explain why the parameter settings in literature [14,38] are used.

Response/Revision: Thanks for comments! We have provided a detailed explanation for the use of parameter settings from references [11, 32] (In lines 350-356). Due to the reorganization of the references, the numbering has changed. Please see the details bellow.

“Literatures [11, 32] were referenced for parameter setting. We referred to existing literature for the following reasons: 1. The parameter settings in the literature are founded on established theories, with authors validating the effectiveness of these parameters. 2. The literature we consulted exhibits a degree of relevance to our research, offering valuable insights for parameter configuration. 3. These parameter configurations yield optimal visual analytical outcomes.In addition, the data has been adjusted repeatedly to achieve a good visual analysis effect.” 

Comment #7 How the uncertainty of the environment is represented in the study.

Response/Revision: Thanks for comments! In this paper, uncertainty is primarily explored during simulations by assigning different initial values. Additionally, the paper includes descriptions of uncertainty and outlines future research directions (In lines 332-338) (In lines 539-543). Please see the details bellow.

“Finally, the uncertainty in the environment in which the supply chain operates may also play a role. Information collaboration benefits depend not only on the amount of information shared and the information absorption and transformation capabilities of both parties but are also influenced by external turbulent environments. However, uncertain environmental factors can have both positive and negative effects, making it difficult to intuitively assess their impact on the system's evolutionary outcomes. Addressing this potential factor is a focus of future research for the author.

Furthermore, the supply chain environment is inherently uncertain, with environmental factors capable of both positive and negative impacts. These uncertainties can significantly affect system evolution outcomes, although discerning their precise influence can prove challenging. Future research endeavors may delve into the realm of uncertainty theory and methods to shed light on environmental uncertainties.”

Comment #8 The authors should analyze the results of the experiment to help readers understand the meaning and results of the experiment, rather than just describing the content of the picture.

Response/Revision: Thanks for comments! In the simulation section, we have added descriptions of the experimental results to assist readers in understanding the significance and outcomes of the experiments (In lines 370-376) (In lines 387-394) (In lines 402-411) (In lines 426-431) (In lines 441-445) (In lines 455-459) (In lines 468-473). Please see the details bellow.

“It can be observed that the higher initial probabilities for suppliers and retailers expedite the rate of convergence toward cooperation (Figs 2b and 2c). This outcome underscores that when the willingness of both parties for initial information sharing is low, it hampers inter-enterprise information sharing. Hence, whether acting as a supplier or retailer, proactive engagement in constructive information exchange with supply chain counterparts, the establishment of trust mechanisms, and the actualization of information sharing are advisable.

Hence, elevated information sharing costs serve as a deterrent to inter-enterprise information sharing. As costs rise, enterprises become less inclined to engage in information sharing. At this juncture, it is imperative to harness advanced technologies such as 5G, artificial intelligence, and the Internet of Things (IoT) to reduce costs. This, in turn, ensures mutual information sharing and fosters supply chain collaboration. In summary, there exists an inverse correlation between the probability of system convergence to sharing and information sharing costs. Lowering information sharing costs and enhancing efficiency can encourage proactive information sharing among decision-makers.

This outcome signifies that when supply chain enterprises anticipate higher additional gains from information sharing, they are more inclined to partake in it. Information sharing within the supply chain grants retailers access to precise market demand information, given their proximity to the market, while suppliers hold inventory, transportation, and logistics data. The exchange of information between both parties further augments market share, thereby increasing additional gains. Consequently, supply chain enterprises should promptly share information conducive to enhancing market share, including market demand forecasts, inventory information, transportation, and logistics data, among others. This sharing, however, should exclude information pertaining to each entity's core competitive advantages.

This underscores that for supply chain enterprises aiming to maximize profits and market share, increasing the quantity and frequency of information sharing is imperative. This information should encompass market demand forecasts, inventory data, transportation, logistics information, and other insights conducive to optimizing decision-making. However, it should exclude details pertaining to each entity's core competitive strengths.

The results emphasize that when supply chain enterprises engage in information sharing, they should strive to mitigate their sensitivity to losses. The focus should remain on long-term gains and development, without undue concern for short-term cost investments. Attention to long-term returns and growth should take precedence over heightened sensitivity to initial costs.

The findings suggest that when supply chain enterprises engage in information sharing and stand to gain from it, they should not solely focus on short-term gains. Instead, they should prioritize long-term gains and development. In situations involving potential gains, they should further invest effectively to maximize returns.

These findings underscore that, when supply chain enterprises engage in information sharing that might result in losses, they should strive to temper their emotional response to such losses. While risk aversion can stimulate the occurrence of information-sharing behaviors, it may inadvertently lead to the neglect of larger potential losses stemming from unforeseen risks. To address this, the implementation of risk-sharing mechanisms can serve as a more effective means of risk management.”

Thank you again for the suggestions and comments, which have further improved our paper.

---

## [Decision Letter · Decision Letter 1]

23 Jan 2024

An Evolutionary Analysis of Supply Chain Collaborative Information Sharing Based on Prospect Theory

PONE-D-23-26347R1

Dear Dr. Hongcheng Gan,

We’re pleased to inform you that your manuscript has been judged scientifically suitable for publication and will be formally accepted for publication once it meets all outstanding technical requirements.

Kind regards,

Bo Huang

Academic Editor

PLOS ONE

Reviewers' comments:

Reviewer's Responses to Questions

**Comments to the Author**

1. If the authors have adequately addressed your comments raised in a previous round of review and you feel that this manuscript is now acceptable for publication, you may indicate that here to bypass the “Comments to the Author” section, enter your conflict of interest statement in the “Confidential to Editor” section, and submit your "Accept" recommendation.

Reviewer #1: All comments have been addressed

2. Is the manuscript technically sound, and do the data support the conclusions?

Reviewer #1: Yes

3. Has the statistical analysis been performed appropriately and rigorously? 

Reviewer #1: Yes

4. Have the authors made all data underlying the findings in their manuscript fully available?

Reviewer #1: Yes

5. Is the manuscript presented in an intelligible fashion and written in standard English?

Reviewer #1: Yes

6. Review Comments to the Author

Reviewer #1: This article is ready for publication.

7. PLOS authors have the option to publish the peer review history of their article (what does this mean?). If published, this will include your full peer review and any attached files.

Reviewer #1: No

---

## [Editor Report · Acceptance letter]

7 Mar 2024

PONE-D-23-26347R1 

PLOS ONE

Dear Dr. Gan, 

I'm pleased to inform you that your manuscript has been deemed suitable for publication in PLOS ONE. Congratulations! Your manuscript is now being handed over to our production team.

Kind regards, 

on behalf of

Professor Bo Huang 

Academic Editor

PLOS ONE